# Molecular and Cellular Factors Associated with Racial Disparity in Breast Cancer

**DOI:** 10.3390/ijms21165936

**Published:** 2020-08-18

**Authors:** Manish Charan, Ajeet K. Verma, Shahid Hussain, Swati Misri, Sanjay Mishra, Sarmila Majumder, Bhuvaneswari Ramaswamy, Dinesh Ahirwar, Ramesh K. Ganju

**Affiliations:** 1Department of Pathology, Ohio State University, Columbus, OH 43210, USA; charan.2@osu.edu (M.C.); Verma.201@osu.edu (A.K.V.); Shahid.sofi@osumc.edu (S.H.); Swati.misri@osumc.edu (S.M.); Sanjay.mishra@osumc.edu (S.M.); 2Comprehensive Cancer Center, Ohio State University, Columbus, OH 43210, USA; sarmila.majumder@osumc.edu (S.M.); Bhuvaneswari.ramaswamy@osumc.edu (B.R.)

**Keywords:** breast cancer, racial disparity, tumor microenvironment, immune cells

## Abstract

Recent studies have demonstrated that racial differences can influence breast cancer incidence and survival rate. African American (AA) women are at two to three fold higher risk for breast cancer than other ethnic groups. AA women with aggressive breast cancers show worse prognoses and higher mortality rates relative to Caucasian (CA) women. Over the last few years, effective treatment strategies have reduced mortality from breast cancer. Unfortunately, the breast cancer mortality rate among AA women remains higher compared to their CA counterparts. The focus of this review is to underscore the racial differences and differential regulation/expression of genetic signatures in CA and AA women with breast cancer. Moreover, immune cell infiltration significantly affects the clinical outcome of breast cancer. Here, we have reviewed recent findings on immune cell recruitment in the tumor microenvironment (TME) and documented its association with breast cancer racial disparity. In addition, we have extensively discussed the role of cytokines, chemokines, and other cell signaling molecules among AA and CA breast cancer patients. Furthermore, we have also reviewed the distinct genetic and epigenetic changes in AA and CA patients. Overall, this review article encompasses various molecular and cellular factors associated with breast cancer disparity that affects mortality and clinical outcome.

## 1. Introduction

Breast cancer is the second most common cancer and a leading cause of cancer related deaths in women around the globe. In 2020, around 276,480 new breast cancer cases are expected to be diagnosed in the United States alone [1]. Breast cancer, like any other type of cancer, is a multifactorial disease, and can be induced by reproductive factors, genetic mutations, biological carcinogens, chemical hazards, environmental factors, and obesity [2,3,4]. Breast cancer is a complex, wide ranging heterozygous disorder with different molecular and clinical subtypes requiring distinct treatment plans. Breast cancer has five molecular subtypes depending on differential gene expression: Luminal A (hormone receptor^+^ and HER2^−^), luminal B (hormone receptor^+^ and HER2^−^/^+^), basal-like or triple-negative breast cancer (TNBC) (hormone receptor^−^ and HER2^−^), HER2 enriched (hormone receptor^−^ and HER2^−^/^+^), normal-like (hormone receptor^+^ and HER2^−^) [5] (Figure 1). Emerging data on breast cancer incidence, overall survival and death rate show significant disparities in different racial groups [6,7]. For example, the breast cancer death rate is 41% greater in African-American (AA) women than in their Caucasian (CA) counterparts [8]. TNBC is a common molecular subtype among women with *BRCA1* mutations; this type of cancer is more common in AA women [9]. Although the incidence rates of breast cancer in CA and AA women are similar, the mortality rate is still much higher in AA women. The higher mortality rate among AA women has led scientists ponder the role of racial differences between the two groups as an underlying cause of mortality. These differences may be attributed to socio-economic status, access to health care, postoperational care, food habits, biological factors and comorbidity. 

In this review, we summarize a number of clinical factors that influence the outcome of racially disparate breast cancer patients. We have scrutinized the racial disparity in terms of tumor microenvironment (TME) composition, genomic aberrations, and cytokines and chemokines secretion. Knowledge of such racially disparate molecules in breast cancer progression will aid in developing novel targeted therapies and improving the clinical outcome of AA breast cancer patients.

## 2. Racial Disparity in the Composition of Breast Cancer TME

The TME between AA and CA women suffering from breast cancer differs significantly, contributing to higher mortality rates in the former population [10,11]. The breast cancer TME comprises different cells including fibroblasts, adipocytes, macrophages, and dendritic cells, and secretes different growth factors and cytokines which regulate the growth and development of tumor cells (Figure 2). A dynamic cross-talk between stromal components and the tumor is indispensable for tumor progression and growth [12].

In AA and CA women with breast cancer, the largest fraction of leukocytes within the TME are macrophages and monocytes, including the M0, M1, and M2 subsets [13,14,15]. Importantly, these subsets of tumor-associated macrophages and monocytes not only support the growth of tumors, but also help in metastasis. AA women with breast cancer are reported to have a higher number of tumor-associated macrophages (TAMs) in the TME than their CA counterparts [15,16,17]. The TME transforms the infiltrating macrophages and converts their phenotype from M1 to M2, which instead of killing cancer cells, become TAMs, supporting tumor growth. These TAMs are protumorigenic, promoting tumor growth and angiogenesis, and further aiding in tumor invasion and metastasis though their secreted factors, thereby contributing to immune evasion of developing tumors [18].

Higher infiltration of TAMs contributes to poor prognoses of breast cancer by supporting uncontrolled cancer progression and widespread metastasis [19]. M2 macrophages proliferate at a higher rate in AA breast cancer patients through different secreted cytokines and chemokines, compared to CA patients [20]. In addition, resting M0 macrophages and T follicular helper cells are slowly recruited to the tumors and alter the TME in AA breast cancer patients, leading to a decrease in overall survival and disease-free state (DFS) [14]. On the other hand, CA women with breast cancer show a higher proportion of tumor supporting M2 macrophages, resting CD4^+^ memory T cells, and mast cells [15]. Moreover, M2 macrophages are essentially protumorigenic, and memory CD4^+^ T cells are known to mediate a direct role in enhancing antitumor immunity, thereby augmenting DFS [14]. Additionally, mast cells are known to play both pro- and anti- tumor roles in the TME, depending on the response of stromal cells [14]. Regulatory T cells (Foxp3^+^ T cells) have been shown to cause immunosuppression, and are reported to promote the enrichment of protumoral proteins. Regulatory T cells increase the expression of surface molecules that prevent the mounting of immune responses against developing tumors [21]. Treg heterogeneity in terms of function and homeostasis makes it hard to reconcile the predictive value of the number of Treg cells in the TME to clinical outcome [22]. The MHC1 metagene displayed higher expression in AA ER^+^ breast cancer patients compared to CA ER^+^ cohort of patients [15]. Additionally, MHC1 genes play an important role in enhancing regulatory functions of T cells. 

Tumor infiltrating lymphocytes (TILs) within the tumor immune microenvironment (TIME) also differ between AA and CA breast cancer patients [23,24,25]. TILs are predominantly comprised of B cells, T cells, and NK cells, playing a crucial role in antitumor immune response [26]. AA women with the Basal/TNBC subtypes are reported to show more regulatory T cells (Tregs), while there are more CD8^+^ lymphocytes only in the luminal subtypes of breast cancer in AA women [27,28]. A higher number CD8^+^ T cells is indicative of a favorable response to neoadjuvant therapy in various molecular subtypes of breast cancer [25,29,30]. Furthermore, immunohistochemical staining of CD8^+^ T cells revealed that a higher percentage of CD8^+^ T cells are recruited in the tumor of AA breast cancer patients compared to CA women with breast cancer, which is suggestive of the mounting of strong immune response [27]. Other groups have also identified differential TIL in TNBC between AA and CA [23,24]. In the early stage, BC (I-II) AAs have significantly higher numbers of TILs, but no difference was observed between ethnicities in stage III-IV TNBC patients. Conversely, a few reports have described minor or no differences between the two races in terms of TILs recruitment [15,31,32]. Also, the distribution of lymphocyte-predominant (>50% TIL), lymphocyte-moderate (10–50% TIL), and lymphocyte-poor (<10% TIL) cases are comparable between races [15,31,32].

Myeloid derived suppressor cells (MDSCs) are a heterogeneous pool of immune cells which are critical for tumor associated immune suppression [33,34,35]. The prime function of these cells in the TME is the suppression of T cells in an antigen-specific or nonantigen-specific fashion [36,37]. These cells are critical determinants in tumor reactive immune cell exhaustion or suppression, and are promising therapeutic targets against various cancers including breast cancer. Despite its significant role in tumor reactive immune cell manipulation, there is no correlation between racial disparity and MDSC recruitment in the TME. However, Apoliprotein E (ApoE), which influences the recruitment of MDSCs, is overexpressed in AA breast cancer patients than CA counterparts [15,38]. The degrees of Apo E signaling and T cell activation are two essential factors for regulating the TME and also important distinguishing factors between AA and CA TNBC from a prognostic standpoint [15]. 

Gene expression profiling of tumor epithelia and stroma of breast cancer from AA and CA patients revealed that genes overexpressed in AA population were related to biological processes that contribute to chemotaxis and angiogenesis [10]. There are a few reports on differences in gene expression levels of immune cells between AA and CA cohorts of breast cancer patients. Interestingly, in a Nigerian populace, the gene signature for cytotoxic cells was low in all subtypes of breast cancer except the basal subtype, while that for fibroblast cells was highest [17]. Biological signaling networks related to chemotaxis in tumor epithelia and neovascularization in the tumor stroma are significantly enriched in AA group compared to CA women with breast cancer.

Overall, the stroma of AA breast cancer patients have higher inflammation and angiogenesis than those of CA patients. Phophoserine phosphatase-like (PSPHL), a gene overexpressed in tumor stroma of AA breast cancer patients, is known to alter the expression of several cytokines and growth factors which play important roles in Extra Cellular Matrix (ECM) remodeling [10,39]. PSPHL is overexpressed in breast cancer and plays a direct role in tumor–stroma crosstalk in the TME. Disparate overexpression of PSPHL between AA and CA populations has also been observed in prostate cancer and endometrial cancer, and may occur in other human cancers as well [40,41]. Other stromal genes dysregulated in AA women with breast cancer include Ras association domain-containing protein 1 (RASSF1A), Retinoic acid receptor beta (Retinoic acid receptor beta), Spermatogenesis associated 18 (SPATA18), and Sons of sevenless drosophila homolog 1 (SOS1). 

Microvessel density is another measure of angiogenesis, and serves as a prognostic marker in various cancers. The microvessel density in breast tumor specimens from different AA women was high compared to that in their CA counterparts. A higher density of macrophages is known to boost angiogenesis and the infiltration of macrophages. Higher infiltration of macrophages in the TME of AA breast cancer patients might cause higher microvessel density in these patients. A higher microvessel density in AA breast cancer patients also leads to poor clinical outcomes [10,42].

### Cytokines and Chemokines

Cytokines released in response to inflammation and immune reaction can function to inhibit or promote cancer. The presence of a differential cytokine response among AA and CA women suffering from breast cancer has been reported in several studies [13,43,44]. Cytokines can alter the TME and provide valuable information to guide therapeutic intervention [45]. Immune cells such as regulatory T cells, NK cells, myeloid cells, and adipose tissue resident macrophages infiltrate tumors and secrete cytokines that help in building an immune suppressive TME [45,46,47]. Depending on the tumor type, immune cells secrete various cytokines and chemokines that aid in the growth of cancer cells and help in immune evasion. The high proportion of TAMs in AA breast cancer patients might be a consequence of increased production of chemotactic chemokines and cytokines in the TME that attract the M2 macrophages [10]. Some of the crucial chemotactic factors secreted by the immune cells that attract macrophages are resistin, CCL2 (MCP-1), vascular endothelial growth factor (VEGF) and M-CSF-1 [48]. Macrophage-derived resistin in the TME further triggers the infiltration of newer macrophages and other immune cells in the protumor TME, as well as aggravating inflammation [49].

AA women show significantly higher levels of IL-6, resistin and IFN- γ secretion than CA women. IL6, secreted from adipocytes into circulation, has been shown to increase breast cancer risk and tumor size [44]. IL-6 is also produced by CD4^+^ Th2 cells that are predominantly involved in dampening antitumor immune responses [50]. IL-6 regulates insulin, resistin and estrogen, and thereby directly affects breast cancer development [45,51]. AA and CA breast cancer patients have resistin and IL-6 as the most differentially-expressed cytokines, with a relatively higher level of expression in AA patients [52,53]. Resistin and IL-6 expression is also positively correlated with serum levels in breast cancer patients. A number of studies have reported the elevated expression of IL-6 in AA breast cancer patients compared to CA patients [44,45,54]. Notably, the CXCL12/CXCR7/CXCR4 axis plays an important role in breast cancer growth and metastasis, but only CXCL12 has been associated with disparate expression. CA breast cancer patients show higher CXCL12 expression than their AA counterparts, and this correlates with poor prognoses [55].

In addition to patient data, a TNBC-derived cell line MDA-MB-468, derived from an AA breast cancer patient, when treated with resistin, has higher growth and aggressiveness compared to MDA-MB-231 cells derived from a CA patient. CD44, which is a marker for stemness, also increases significantly in resistin-treated MDA-MB-468 compared to MDA-MB-231 [52]. Resistin promotes the growth and aggressiveness of breast cancer cells through STAT3 activation, indicating a potential role of resistin, IL-6 and STAT3 in breast cancer racial disparity in AA women [45].

VEGF and syndecan are widely recognized as angiogenesis related signaling molecules and are reported to be overexpressed in AA compared to CA breast cancer patients [10]. Adipocyte derived pro-inflammatory cytokines such as IL-6, leptin and TNF-α, along with angiogenic factors, not only help in the development of breast tumors, but also promote more aggressive phenotypes. Higher levels of leptin caused by insulin secretion lead to the creation of an autocrine feedback loop which increases mitogenesis and decreases apoptosis in breast cancer cells [56]. Leptin also induces the secretion of pro-inflammatory cytokines like IL-6, TNF-α, IL-2 and IFN-γ [57]. Obese patients being studied for serum adiponectin (corrected for BMI) showed higher levels of IL-6 and C-reactive protein [58]. IL-6, along wih resistin and other cytokines, could be related to higher aggressiveness of TNBC in AA women. Unchecked growth of adipocytes might cause the release of monocyte chemoattractant protein-1 (MCP-1), which causes macrophage infiltration and the activation of resident macrophages [19,42].

Atypical chemokine receptor 1 (ACKR1) plays a pivotal role in immune regulation. AA women have a higher proportion of tumors that are ACKR1 negative compared with CA women [59]. ACKR1 positive tumors differ from ACKR1 negative tumors in their immune responses. ACKR1 expression in tumors is correlated with higher pro-inflammatory chemokines, i.e., CCL2/MCP-1. ACKR1 alleles specifically expressed in AA women likely drive these correlations, which help in the overall and relapse-free survival of patients with tumors showing higher expression of ACKR1 [59]. CCL7 is more elevated in AA women with breast cancer than in CA patients. CCL7 binds to CCR1, CCR2 and CCR3 and activates MAPK signaling, leading to EMT and TAM recruitment from endothelial leakiness [31,60]. Additionally, CCL7, CCL8 and CCL5 are also elevated in AA patients with TNBC [15,61]. CCL5 promotes breast cancer in a p53-dependent manner through CCR5, and antagonizing the CCL5 receptor inhibits CCR5-mediated angiogenesis [62,63]. Higher expression levels of CCL17 and CCL25 in breast cancer patients have been attributed to poor overall survival in the AA population, but no such correlation has been observed in the CA population. In addition, higher expression of CCL8 decreases overall survival only in CA breast cancer patients. However, CCL25 has served as an indicator of poor prognosis in AA breast cancer patients; its expression was correlated with improved overall survival in CA breast cancer patients [61].

Higher expression of CCL7, CCL11 and CCL20 in AA breast cancer patients has been shown to correlate with higher overall survival and better prognoses. Together, CCL17 and CCL25 in AA breast cancer patients decrease overall survival, while high CCL8 decreases overall survival in CA patients [61,64]. CCL7, CCL17, CCL20 and CCL25 are significantly more elevated in AA breast cancer patients compared to CA patients. Slightly higher expression of CCL8 and CCL7 has also been reported in AA tissues [61,64]. At gene level, AA women with breast cancer showed higher expression of several key cell cycle regulating genes, including CCNE2, CCNB1, CCNA1, CDKN2A and other tumor related genes CRYBB2, TMPO, AMFR, PSPHL, which directly impact the development and aggressiveness of tumors. A higher expression of interferon has been observed in AA breast cancer patients, indicating their ability to better respond to immunotherapy [15,17].

Tumor stroma also contributes to the expression of chemokines including CXCL10 and CXCL11 and other stromal protein PSPHL. CXCL10 and CXCL11 chemokines are canonical ligands for the CXCR3 receptor [10]. *CXCL10, CXCL11* and *ISG20* are interferon γ-regulated genes which affect the expression of a number of other genes in ER(^+^) and ER(^−^) breast tumors. In ER(^+^) tumors, the presence of an interferon gene signature is an indicator of estrogen-mediated host immunity, and is involved in tumor development, growth, survival and metastasis [65,66,67]. In ER(^−^) tumors, HLA-D family members *HLA-DQA1* and *HLA-DQB1* were the most differentially expressed at both the mRNA and protein levels [10]. The disparate distribution of different immune cells, cytokines and chemokines among AA and CA breast cancer patients has been catalogued for future experimental designs and the generation of hypotheses (Table 1 and Table 2).

## 3. Influence of Genetic and Epigenetic Factors on Breast Cancer Disparity

Several studies have linked the differential expression of genes with downstream biological events that differ by ethnicity/race to poorer prognoses of the disease, but only a few have experimentally validated these correlations [68,69,70]. Antibody-based detection on a microarray identified several proteins that were differentially expressed in AA and CA breast cancer patients. Noninvasive, race-specific serum-based biomarkers are helpful in understanding why the burden among AA breast cancer patients is higher than among their CA counterparts. Three race-specific protein markers, i.e., VEGFR2, c-Kit, and Retinoblastoma (Rb), are overrepresented in tumors of AA breast cancer patients [71]. VEGFR2 protein is reported to affect breast cancer metastasis, prognosis and racial disparity [72,73]. High expression of *VEGFR2* could be exploited as a therapeutic target against breast cancer in AA patients. c-kit is overexpressed in AA breast cancer patients and is involved in the growth and survival of *BRCA1* mutated cells [71]. Lower serum levels of Rb protein were detected in AA breast cancer patients than other racial groups [71]. Rb acts as a checkpoint molecule between the G1 to S transition of the cell cycle, preventing the unchecked proliferation of potential tumor cells. Lower Rb expression causes unchecked proliferation of tumor cells [74]. In addition, CLCA2, modulated by p53 in response to DNA damage stimuli serves as a prognostic marker for TNBC only in AA patients [75]. A detailed analysis of these factors in terms of their effect on clinical outcome in racially disparate groups is presented below. 

### 3.1. Signaling Molecules Associated with Cellular Growth

Cancer is the cumulative effect of a multistep processes and multiple mutations rather than a single gene event. The biological network of microdissected tumor epithelia and tumor stroma has been extensively studied and is believed to affect overall survival in breast cancer. ER-negative, high grade breast tumors of younger AA women showed increased expression of cyclin E [76]. Similarly, high grade, advanced tumors showed increased cyclin B expression [77], and breast tumors from AA women demonstrated increased expression of cyclin B [10]. Cyclin B is essential for mitosis and G2-M transition during the cell cycle. TNBC tumors have irregularities in the cell cycle gene expressions (high expression of p16, p53, and cyclin E, and low cyclin D1 expression) which might contribute to phenotypic changes in tumors of AA and CA women [43,78,79]. p16 binds with CDK4/6 in order to inhibit cyclin D binding, and its overexpression has been reported in tumors derived from AA breast cancer patients, in contrast to those from CA women [10,78,80].

Lactotransferrin (LTF) has been reported to show disparate expression among AA and CA breast cancer patients. It showed a more than eight-fold expression difference between an AA and CA cohort of breast cancer patients. It is implicated in a variety of functions including cellular growth, differentiation, inflammation, and in regulating immune response [81]. It also affects the MAPK and Akt signaling pathways and induces senescence or growth arrest [82,83]. The C4BPA gene encodes the alpha chain of the C4b-binding protein which inhibits the complement cascade; it has greater than six-fold greater expression in AA women [84]. Monoclonal antibodies have the ability to induce the complement system and stimulate complement-dependent cytotoxicity (CDC) that results in tumor cell clearance [85]. Higher C4BPA expression impedes the complement system and CDC, thereby helping in tumor cell immune escape and survival.

Tumor suppressor gene p53 is a well-known for its role in DNA repair and inducing apoptosis [86,87]. In most human carcinomas, p53 gets mutated, and its mutational status is also an independent prognostic marker of AA breast cancer [88,89]. Breast cancer mitogen insulin-like growth factor 1 (IGF-1) levels increase in AA women after multiple pregnancies and promote breast cancer progression [90]. IGF-1 also enhances cell cycle progression in G1/S checkpoint-compromised cells. Metalloproteases such as ADAMTS15 have been reported to inhibit breast cancer cell migration [91], and their reduced expression could accelerate breast cancer progression in AA women.

Recently, serum-derived exosomes have gained interest in the context of addressing racial disparity. Exosomes play a variety of important roles in breast cancer including tumor growth, metastasis, immunosuppression, and drug resistance [92]. Annexin 2 (ANX2) had been associated with angiogenesis and ECM modification [93,94]. Serum-derived exosomes from AA women TNBC patients exhibited higher ANX2 expression compared to those from CA women. ANX2 is associated with the aggressiveness of breast cancer and is considered as a molecular marker for different breast cancer subtypes [95]. In addition, an overexpressed ANX2 AA cohort was also associated with poor overall survival and poor disease-free survival [96].

Martin et. al. (2009) observed more than 400 differentially expressed genes in AA and CA populations with breast tumors [10]. An in silico data analysis of breast tumor patient samples revealed that ACTL8 and PGLYRP1 were differentially expressed between the AA and CA groups [97]. In addition, DNAJB8, a member of the heat shock protein (HSP) family that plays a key role in protein folding, showed higher expression only in AA samples [98].

AA breast cancer patients also show increased expression of STAT1 (which acts as a tumor suppressor) in the early stages of tumor initiation [99]. Crystallin β B2 (CRYβB2), a protein, constitutes a predominant fraction of vertebrate eye lenses and has been highlighted for its correlation in overall survival of AA population in a number of cancers. Both CRYβB2 and CRYβB2P1 are aberrantly expressed in AA breast cancer patients, but these genes stimulate tumor progression independently [8]. Together, these genes can be further tested experimentally in order to improve the disparate clinical outcome of AA breast cancer patients.

### 3.2. Gene Mutations

Genetic mutations, which are spontaneous or derived from any external factor like chemical exposure, UV/ion radiation exposure or hereditary factors, can also increase the risk of breast cancer. There are two types of mutations, i.e., genetic and somatic mutations. A germline variant is a change or mutation in a gene that is inherited from the parents. Somatic mutations occur due to genetic and environmental exposure, and cause different types of cancers. Mutations in various genes contribute to a high risk of breast cancer. Three well-known genes, *BRCA1, BRCA2*, and *PALB2*, often get mutated and increase the risk for breast and/or ovarian cancer [100]. Abnormal expression of *BRCA1, BRCA2*, and *PALB2* has been observed in about 10% of breast cancer cases [101]. BRCA1 and BRCA2 act as tumor suppressor proteins, and patients with inherited mutations therein are more likely to develop aggressive breast cancers [102,103,104]. These proteins assist in DNA repair, and therefore, play an important role in the fidelity and stability of genetic material. Mutations in *BRCA1* or *BRCA2* can alter the function of these proteins, and the process of DNA damage repair is hampered. Therefore, cells are more likely to develop additional genetic alterations that can lead to the development of cancer. About 5–10% of breast cancers are caused by a gene mutation in *BRCA1* and *BRCA2* [105]. *BRCA1* gene mutation in women accelerates breast cancer at a younger age [106]. Individuals who carry mutations in either their *BRCA1* or *BRCA2* genes can pass them on to 50% of their progeny.

Some *BRCA1* and *BRCA2* mutations that are inherited increase breast cancer risk in women. Together, mutations in the *BRCA1* and *BRCA2* genes contribute to about 20–25% of inherited breast cancers [107]. In addition, genetic testing of breast cancer patients revealed that the population which is of African or Bahaman decent showed higher frequencies of *BRCA1* and *BRCA2* mutations [108]. AA women carrying mutations in the *PALB2, RAD51C,* and *RAD51D* genes are more susceptible to ER-negative breast cancer [107]. Like *BRCA1* and *BRCA2* gene mutations, *PALB2* mutations are also associated with high risk of breast cancer. PALB2 is a tumor suppressor protein which interacts with BRCA1 and BRCA2 to repair DNA damage and breaks [109]. Apart from *BRCA1* and *BRCA2* gene mutations, *ATM, CDH1, CHEK2, PALB2, PTEN, STK11,* and *TP53* mutations potentially increase breast cancer risk [110]. The top five genes most commonly mutated in AA and CA breast tumors are *TP53, PI3KCA, GATA3, CDH1,* and *MLLT3* (*n =* 663, CA and *n =* 105 AA breast tumor samples). However, of these five genes, the mutation frequency of two genes, namely *TP53* and *PI3KCA*, differs between CA and AA women. A higher percentage of AA women with breast cancer harbor *TP53* mutations (42.9% AA vs 27.6% CA), whereas PI3K mutations were less commonly observed in AA than in CA women (20% AA vs 33.9% CA). Further, the risk of tumor recurrence is also higher in the AA than in the CA population. Such racial disparity in terms of tumor relapse is attributed to a breast cancer subtype, i.e., TNBC and the occurrence of *TP53* mutations [7].

*TP53* gene mutations are observed in about 50% of all breast cancers. There are more than 2500 different types of mutations documented in p53. p53 mutations are commonly found in AA cohorts of breast cancer patients, and they affects overall survival [111,112]. Shiao and colleagues (1995) observed more G:C to A:T transitions at non-CpG sites in black women compared to white; this phenomenon contributes to poor survival in AA breast cancer patients [111]. p53 status may predict survival independently with adjusting stage, tumor grade, and subtype, which is useful to identify AA women in the high risk category for breast cancer mortality [88]. Some genes get mutated occasionally, including *BRIP1, MLH1, MLH2, MRE11A, NBN, PALB2, PTEN, RAD50, RAD51C,* and *SEC29B* [113]. Mutations in the *CHEK2, ATM, ERCC3*, and *FANCC* genes are linked to a moderate risk of ER-positive breast cancer. RECQL gene mutations present a moderate risk of all types of cancer [107]. Both *CHK1* and *CHK2* have important functions in cell cycle regulation, and have been found to be mutated more frequently in AA breast cancer patients [114]. Additionally, breast cancer patients predisposed to somatic mutations like *CHEK2, ATM, BARDI, PALB2, RAD51C, TP53, PTEN*, *MLH1* and *MLH2* are also at higher risk of developing breast cancer [108].

### 3.3. Genomic Alterations

Copy number alterations are frequent in breast cancer. The identification of copy number alterations specific to a breast cancer subtype defines the mechanism of disease initiation and progression. Chromosomes 1q and 8q show high gain events, while chromosomes 8p, 10q, 11q, 12q, and 16q have a higher frequency of loss events [115]. Loo et. al. (2011) observed higher frequencies of gain and loss events in breast tumors of AA women than CA women [116]. The frequency of gain of copy number in the 13q31-13q34 chromosomal region was observed to be twice as high in TNBC from AA women than from CA women. Previously, Melchor et al. (2009) reported the close association of 13q31-13q34 chromosomal region amplification with TNBC [117].

Cullin4A (CUL4A) and transcription factor Dp-1 (TFDP1) at the 13q34 chromosomal region show increased protein expression in breast tumors. The overexpression of CUL4A and TFDP1 are associated with shorter overall and disease-free survival in breast cancer [118,119]. In human breast cancer, genetic deletions have been demonstrated to be among major genetic abnormalities. The loss of heterozygosity in breast cancer has been documented at several chromosomal locations, including 1p, 1q, 2p, 3p, 6q, 7q, 8q, 9q, 11p, 11q, 13q, 15q, 16q, 17p, 17q, 18p, 18q, and 22q [120]. The loss of heterozygosity in a few chromosomal regions, or at locations such as p53 at chromosome 17q13 [121], BRCA1 at 17q21 [122] and BRCA2 at 13q12-13 [123], has been shown to influence breast cancer outcome.

### 3.4. DNA Methylation

Epigenetic changes are known to play important roles in various cancers. Racial disparity in molecular epigenetic markers between AA and CA breast cancer patients has also been documented. Epigenetic technological advancements have played a crucial role in improving the clinical outcome of various human cancers. DNA methylation is a very common epigenetic phenomenon. Tumor suppressor genes like *p16, BRCA1, GSTP1, TIMP-4*, and *CDH1* contribute to breast cancer progression and growth. The promoter regions of these genes were frequently observed to be methylated [124]. Racial differences are closely associated with altered DNA methylation in breast tumors [125,126,127,128].

Although very little is known about the causal factors of hypermethylation, it is hypothesized that these events strongly influence breast cancer progression. There are a few reports suggesting that a low level of folate in breast tissue and increased alcohol consumption can result in hypermethylation of p16 gene [129]. Mehrotra et al. (2004) [130] reported that epigenetic modification, like DNA methylation, among ER-negative tumors of AA women below 50 years of age showed higher cyclin D2 promoter methylation than in CA women. Similarly, *RASSF1A*, a tumor suppressor gene that controls numerous checkpoints of cell cycle and apoptotic pathways [131], was methylated more in AA than in CA breast cancer patients [130]. Additionally, the *RARβ* and *HIN-1* genes were also frequently found to be methylated in breast tumors of AA women [130].

## 4. Conclusions

In this review article, we have comprehensively discussed the molecular pathways that promote breast cancer incidence and mortality among AA patients. Breast cancer results in higher mortality among AA women than among their CA counterparts. Moreover, AA women are at higher risk of developing more aggressive breast tumors, even at young ages, than white women. Recently, TME has been shown to be a potential therapeutic target against solid tumors including breast cancer. Emerging data on the distinct TME composition between AA and CA breast cancer patients warrants further study in order to develop the TME as a novel therapeutic target and thereby improve clinical outcomes, especially among AA breast cancer patients. Furthermore, the differential expression of cytokines and chemokines also significantly affects the clinical outcome of AA breast cancer patients, and could be used as a potential prognostic marker and therapeutic target. In addition, genetic mutations also influence the clinical outcome of AA breast cancer patients. Finally, racially disparate epigenetic modifications have recently been reported in breast cancer, demanding further investigations to improve the therapeutic strategies and clinical outcome of the disease. Nonetheless, available data on the composition of TME and molecular/cellular changes in terms of gene and protein alterations in breast cancer patients show a close association with racial differences. Overall, our detailed analysis will help in designing novel treatment strategies to improve the survival and quality of life among breast cancer patients.

## Figures and Tables

**Figure 1 ijms-21-05936-f001:**
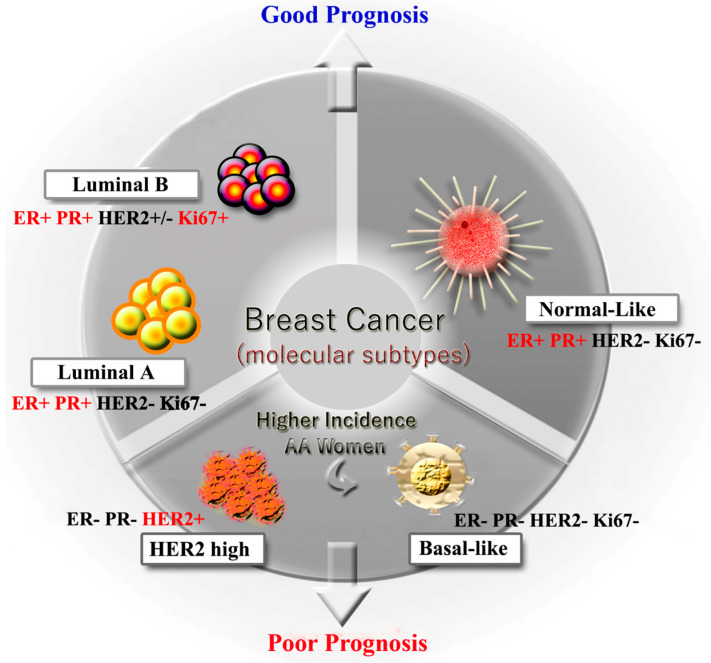
Different molecular subtypes of breast cancer.

**Figure 2 ijms-21-05936-f002:**
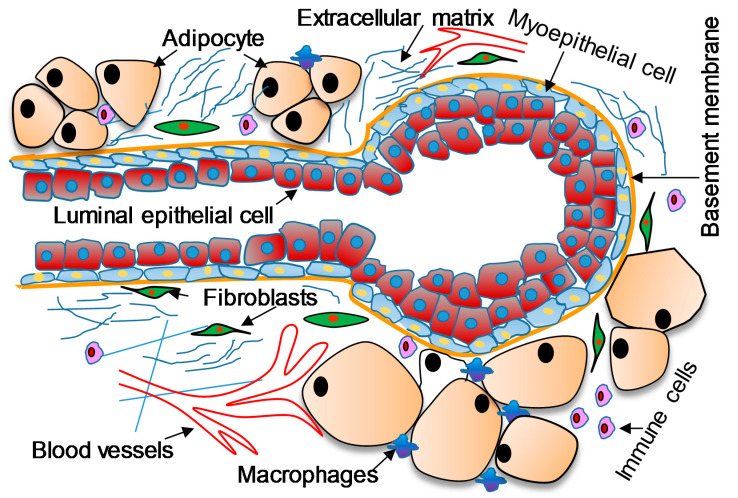
The complex tumor microenvironment of breast cancer.

**Table 1 ijms-21-05936-t001:** Disparate regulation of different cells in breast cancer.

Different Cells	Function	African-American	Caucasian-American	References
Regulatory T cells	Immune suppression	High	Low	[28]
CD8^+^ T cells	Antitumor	High	Low	[27]
Tumor-associated macrophages	Pro- and Anti-tumor	High	Low	[10,16,17]
Resting M0 macrophages	Immune suppression	High	Low	[15]
Follicular T_H_ cells	Immune suppression	High	Low	[15]
M2 macrophages	Protumor	Low	High	[15]
Resting CD4^+^ T cells	Antitumor	Low	High	[15]
Mast cells	Tumor progression	Low	High	[15]
Monocytes	Tumor progression	Low	High	[15]
Fibroblasts	Tumor growth	High	Low	[17]

**Table 2 ijms-21-05936-t002:** Disparate recruitment of proteins and cytokines in breast cancer.

Proteins	Function	African-American	Caucasian-American	References
VEGF	Angiogenesis; Protumor	High	Low	[10]
IL6	Protumor	High	Low	[44]
Resistin	Increased tumor growth; metastasis	High	Low	[44]
IFN-γ	Antitumor	High	Low	[44]
Syndecan	Protumor; Invasiveness	High	Low	[10]
CCL7	Protumor; Increased proliferation	High	Low	[64]
CCL8	Protumor; Growth and metastasis	High	Low	[64]
CCL5	Protumor; macrophage recruitment, collagen deposition	High	Low	[64]
CCL17	Protumor; cell migration and invasion	High; Poor survival	High: Improved survival	[64]
CCL25	Protumor; Migration, invasion	High; Poor survival	High: Improved survival	[64]
CCL8	Protumor; metastasis, Chemotaxis,	No effect	High: poor survival	[64]
CCL7	Protumor, Angiogenesis, Chemotaxis	High: increase survival	No effect	[64]
CCL11	Protumor; Immune suppression	High: increase survival	No effect	[64]
CCL20	Protumor; drug resistance, migration, proliferation	High: increase survival	No effect	[64]
CXCL12	Tumor growth, and metastasis	–	High Poor prognosis	[55]

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
