# Peer review of "Molecular and Cellular Factors Associated with Racial Disparity in Breast Cancer"

_ijms, 2020, doi:10.3390/ijms21165936_

Round 1

Reviewer 1 Report

The authors have nicely written the review. There are just a few minor concerns that they have to address before acceptance. Please see below.

  1. If possible, please increase the font size of the text in figure 1.
  2. On line 88- ref (12); line 114- ref (21,30,31); and line 239- ref (21,33); are highlitted. please remove.
  3. Please correct the heading numbering starting from the Introduction.
  4. Table 1- arrange the refernce column.

Author Response

We would like to thank the reviewer for providing insightful comments and suggestions on our manuscript. Here, we have attached a point by point response.

Response to Reviewer 1 Comments

  1. If possible, please increase the font size of the text in figure 1.

Response: When we increase the font size of figure 1, it looked disproportional. Therefore, we suggest to keep the figure in its present form.

  1. On line 88- ref (12); line 114- ref (21,30,31); and line

Response: Corrected in the manuscript.

  1. 239- ref (21,33); are highlighted. please remove.

Response: Corrected in the manuscript.

  1. Please correct the heading numbering starting from the Introduction.

Response: Corrected in the manuscript. 

  1. Table 1- arrange the reference column.

Response: Corrected in the manuscript.

Reviewer 2 Report

Manish Charan, et al broadly summarized the current knowledge about the disparity of breast cancer in molecular factors and genetic alterations between African and Caucasian patients. It will be helpful if adding the exosomes diversity between AA and CA as exosome is widely documented by recent publications to play a crucial role in TME. In addition, CXCL12/CXCR7/CXCR4 axis is involved in TME and metastasis of breast cancer. The authors need to discuss it if related studies have revealed the difference between the two populations.

Author Response

We would like to thank the reviewer for providing insightful comments and suggestions on our manuscript. Here, we have attached a point by point response.

Response to Reviewer 2 Comments

It will be helpful if adding the exosomes diversity between AA and CA as exosome is widely documented by recent publications to play a crucial role in TME. In addition, CXCL12/CXCR7/CXCR4 axis is involved in TME and metastasis of breast cancer. The authors need to discuss it if related studies have revealed the difference between the two populations.

Response: We have incorporated a section regarding exosome and racial differences in the text. However, we did not see any report on presenting a correlation between racial disparity and CXCL12/CXCR7/CXCR4 signaling axis in the TME. However, CXCL12 has higher expression among CA breast cancer patients; we have added this observation in the manuscript.

Reviewer 3 Report

Review of the Article by CHaran et. al “Molecular and Cellular Factors Associated with Racial Disparity in Breast Cancer”

It is certainly an interesting topic, and warrants investigation. The authors have provided a lot of information on the various molecular and cellular factors at play in breast cancer.

The abstract fails to give the reader a sense of direction of this review article, it seems as if the authors are not clear on what the outcome of this review is.

The language used in the introduction needs revision, consider revising the writing style. Sentences are not cohesive and mere standalone statements, often referencing is limited and could contain more succinct information. While the topic has relevance, the introduction does not provide a robust argument for this review. Authors have not even considered socio-economic factors relating to this large discrepancy. There are other factors at play, the author must acknowledge.

Some of the statements made in the article are not based on strong papers to support the claims made by the author (for example line 72-74).

Very little evidence is included on the studies the authors refer to, are these robust studies with large numbers, did the authors investigate the studies for bias, how statistically sound are the included studies, this will have an impact on the review.

The authors fail to pull all this information together to describe the outcome of the review to the reader.

Minor

  • Line 31 remove the web link it should be referenced in line with standard referencing guidelines
  • No reference here for “Tumor infiltrating lymphocytes (TILs) within the tumor immune microenvironment (TIME) also differ between AA and CA breast cancer patients.” (line 101-102)
  • Keep formatting same, in line 141 authors start sentence with brackets (Phophoserine phosphatase-like) PSPHL and in line 143 Extra Cellular Matrix (ECM). Ideally sentences should not start with a bracket.
  • Authors state that this has been reported in multiple reports, however the reference just highlights one paper. “Presence of a differential cytokine response among AA and CA women suffering from breast cancer has been reported in several studies (39).”
  • Table 1, the formatting needs to be reviewed, the text alignment is off. It would be worth including separate tables for proteins and immune cells for clarity and briefly describing key functions of those in the table for context.
  • Line 254-256 “Several studies have linked differential expression of genes with downstream biological events that differ by ethnicity/race and may contribute to poorer prognosis of the disease but only a few studies are performed to experimentally validate these correlations.” No references provided!

Author Response

We would like to thank the reviewer for providing insightful comments and suggestions on our manuscript. Here, we have attached a point by point response.

Response to Reviewer 3 Comments

While the topic has relevance, the introduction does not provide a robust argument for this review. Authors have not even considered socio-economic factors relating to this large discrepancy.

Response: Our manuscript is mainly focusing on the disparity between molecular and cellular factors in breast cancer. Although, we have mentioned the existing socioeconomic disparity in the Introduction part of the manuscript.

There are other factors at play, the author must acknowledge. Some of the statements made in the article are not based on strong papers to support the claims made by the author (for example line 72-74).

Response: Only a few reports have published regarding immune cell regulation and breast cancer racial disparity. Therefore, in order to cover more information on the subject, we had to include research articles and abstracts from these papers.

Very little evidence is included on the studies the authors refer to, are these robust studies with large numbers, did the authors investigate the studies for bias, how statistically sound are the included studies, this will have an impact on the review.

Minor

  • Line 31 remove the web link it should be referenced in line with standard referencing guidelines

Response: Corrected in the manuscript.

  • No reference here for “Tumor infiltrating lymphocytes (TILs) within the tumor immune microenvironment (TIME) also differ between AA and CA breast cancer patients.” (line 101-102)

Response: Corrected in the manuscript.

  • Keep formatting same, in line 141 authors start sentence with brackets (Phophoserine phosphatase-like) PSPHL and in line 143 Extra Cellular Matrix (ECM). Ideally sentences should not start with a bracket.

Response: Corrected in the manuscript.

  • Authors state that this has been reported in multiple reports, however the reference just highlights one paper. “Presence of a differential cytokine response among AA and CA women suffering from breast cancer has been reported in several studies (39).”

Response: Updated the statement with more references in the manuscript.

  • Table 1, the formatting needs to be reviewed, the text alignment is off. It would be worth including separate tables for proteins and immune cells for clarity and briefly describing key functions of those in the table for context.

Response: We have added another column for function and made two separate tables as suggested.

  • Line 254-256 “Several studies have linked differential expression of genes with downstream biological events that differ by ethnicity/race and may contribute to poorer prognosis of the disease but only a few studies are performed to experimentally validate these correlations.” No references provided!

Response: Reference is added in the manuscript.

Round 2

Reviewer 3 Report

Thank you for your response, the article has been updated to a standard that is satisfactory and publication of this article "Molecular and Cellular Factors Associated with Racial Disparity in Breast Cancer".

Author Response

We would like to thank the reviewer for providing insightful comments. Here, we have addressed the raised concerns point by point.

Comments:

While the topic has relevance, the introduction does not provide a robust argument for this review. Authors have not even considered socio-economic factors relating to this large discrepancy.

Response: Our manuscript is mainly focusing on the disparity between molecular and cellular factors in breast cancer. Although, we have mentioned the existing socioeconomic disparity in the Introduction part of the manuscript.

There are other factors at play, the author must acknowledge. Some of the statements made in the article are not based on strong papers to support the claims made by the author (for example line 72-74).

Response: Only a few reports have published regarding immune cell regulation and breast cancer racial disparity. Therefore, in order to cover more information on the subject, we had to include research articles and abstracts from these papers.

Very little evidence is included on the studies the authors refer to, are these robust studies with large numbers, did the authors investigate the studies for bias, how statistically sound are the included studies, this will have an impact on the review.

Response: We have included latest and good quality papers for this review.  These papers have taken sample size and bias into account, we believe all references highlights key racial differences and associated breast cancer clinical outcome.

Minor

  • Line 31 remove the web link it should be referenced in line with standard referencing guidelines

Response: Corrected and highlighted in the manuscript.

  • No reference here for “Tumor infiltrating lymphocytes (TILs) within the tumor immune microenvironment (TIME) also differ between AA and CA breast cancer patients.” (line 101-102)

Response: Corrected and highlighted in the manuscript.

  • Keep formatting same, in line 141 authors start sentence with brackets (Phophoserine phosphatase-like) PSPHL and in line 143 Extra Cellular Matrix (ECM). Ideally sentences should not start with a bracket.

Response: Corrected and highlighted in the manuscript.

  • Authors state that this has been reported in multiple reports, however the reference just highlights one paper. “Presence of a differential cytokine response among AA and CA women suffering from breast cancer has been reported in several studies (39).”

Response: Updated the statement and highlighted with more references in the manuscript.

  • Table 1, the formatting needs to be reviewed, the text alignment is off. It would be worth including separate tables for proteins and immune cells for clarity and briefly describing key functions of those in the table for context.

Response: We have added another column for function and made two separate tables as suggested.

  • Line 254-256 “Several studies have linked differential expression of genes with downstream biological events that differ by ethnicity/race and may contribute to poorer prognosis of the disease but only a few studies are performed to experimentally validate these correlations.” No references provided!

Response: Reference is added and highlighted in the manuscript.

We have attached the corrected manuscript for their review.

Thanks,

Dinesh Ahirwar